# WHO target product profile for TB detection at peripheral settings: 2024 update

**Mikashmi Kohli**[ID]1*, **Alexei Korobitsyn**2©, **Nazir Ismail**2©, **Matteo Zignol**2, **Tereza Kasaeva**[ID]2, **Puneet Dewan**[ID]3, **Morten Ruhwald**1, **the Scientific TPP Development group**¶

1 FIND, Geneva, Switzerland, 2 Global TB Department, WHO, Geneva, Switzerland, 3 Bill &Melinda Gates Foundation, Seattle, Washington, United States of America

¶ Members of the Scientific TPP Development group is provided in the Acknowledgements.
© These authos are contributed equally to this work.
\* Mikashmi.kohli@finddx.org

## Abstract

In 2014, the World Health Organization (WHO) published target product profiles (TPPs) which detailed minimal and optimal criteria to address high-priority TB diagnostic needs. Since then, the TB community's needs have evolved; there has been a surge in new and innovative platforms suggesting use of alternative non-invasive specimens for TB diagnosis. This updated TPP used evidence-based modelling to guide the decision-making process for specific characteristics which was a novel approach to this process. This document focusses on point of care, near point of care and low complexity assays using sputum and non-sputum-based specimens. The standard WHO protocol was followed for this TPP document including Delphi process, public comments and TPP development group consultations. Our modelling work suggests that lower accuracy estimates for point of care, easily accessible tests are acceptable to achieve comparable or better case detection than the current standard of care. In this document, we describe the process of updating the TPP for TB diagnostic tests at peripheral settings, highlight key updates, use of modelling to inform this update, and discuss guidance regarding technical and operational specifications.

## Introduction

Tuberculosis (TB) continues to be a major cause of morbidity and mortality globally despite being curable and preventable. In 2022, over 10 million people suffered from TB disease of whom an estimated 1.6 million people died from it [1]. The diagnostic gap for TB increased to 4.2 million during the COVID-19 pandemic [2] which has recovered to the pre-pandemic levels in 2022 of over 3 million. WHO-recommended rapid diagnostics (WRD) are a major advancement in the management of TB. However, uptake has been slow, with only 47% of all notified cases tested in 2022 and

**Data availability statement:** All data has been provided as part of this manuscript.

**Funding:** MK, MR received funding for this work through the Bill & Melinda Gates Foundation (INV-045721). PD is an employee of the Gates Foundation and was involved in the conceptualization of this work. The funders had no role in study design, data collection and analysis, decision to publish, or preparation of the manuscript.

**Competing interests:** The authors have declared that no competing interests exist.

about one-third of all TB diagnostic testing sites globally having access to these tests. In 2023, the WHO issued a standard on universal access to rapid TB diagnostics [3] and specifies twelve benchmarks across the diagnostic cascade to be tracked. Emphasis is placed on reaching all individuals in need of testing, providing primary healthcare access (PHC) to testing with timely and quality-assured services and achieving universal drug susceptibility testing (DST).

Adopting current WRDs, while important, will be insufficient on its own to fully achieve the standard. Technologies that are fit for purpose at the PHC level and the use of alternative sample types will likely be critical going forward, and this point is well articulated by Pai and colleagues on the transitions that are needed [4]. The diagnostic landscape for TB diagnosis has been rapidly evolving, with innovative sampling methodologies using tongue swabs or urine and molecular diagnostic platforms that are low-cost and could be deployed at the PHC level. Furthermore, the COVID-19 pandemic led to the development of many rapid and innovative technologies which have the potential to address TB diagnostic needs. To support new product development for TB detection at the peripheral level of healthcare, the WHO updated the target product profile and defined product characteristics to be considered when developing and investigating new tools for TB detection.

This TPP was an update to the 2014 series of TB TPPs [5] and replaces the previous two TPPs on biomarker-based non-sputum and rapid sputum-based testing. The TPP on DST was updated in 2021 [6] and is out of scope for this TPP, while the community-based triage test will now feature separately in a specific TPP on tests for TB screening. Much has changed since 2014, and an update to the TB detection TPP was needed, considering the surge in new sample types, new portable instruments, innovative sampling strategies, and the capture of the full spectrum of TB disease, including subclinical TB. All WHO TPPs are regularly updated on the WHO TPP directory [7].

This updated TPP used evidence-based modelling to guide the decision-making process for specific characteristics. This was considered important to complement the expert opinion-based process usually used and could aid the decision-makers in better understanding the implications and trade-offs when planning.

The target audience for these TPPs includes commercial test developers and manufacturers, members of academia and research institutions, research funding agencies, regulatory agencies, nongovernmental organizations and private sector implementers, National TB programmes (NTPs), civil society organizations, and donors.

## Methodology

### A. Stakeholder engagement

WHO constituted a Scientific TPP Development Group (STDG), consisting of subject matter experts, public health specialists, country policymakers, donors, regulators and civil society representatives. Members of the STDG were engaged throughout the TPP development process and proposed the final TPPs.

## B. Definitions for TPP document

a. <u>Specifications:</u> Following the standard WHO format, each characteristic of the TPP document has a "minimal" and an "optimal" specification. Detailed definitions of these have been published before [6]. Although it is expected that potential diagnostic products would meet all the required minimum criteria of the "Target Product Profile TB diagnostic tests for peripheral settings", and as many of the optimal requirements as possible, potential trade-offs on performance, cost, impact, and operational characteristics would need to be considered for WHO policy; thus, the criteria are indicative rather than absolute.

b. <u>Healthcare settings:</u> three types of tests are defined based on the healthcare setting where tests will be performed and the complexity of the test, including the infrastructure, equipment and skills required (Fig 1). Implicit is the accessibility and timeliness in linking a person to treatment decisions for people presumed to have TB.

1. **Point of care tests (PoC)**: These tests would be instrument-free, not requiring any particular infrastructure in terms of electricity, equipment, or cold chain and can be placed in healthcare settings without laboratories. No special skills are needed to perform this test. An example is a dipstick or lateral flow test). It could have small ancillary devices which can be used at PoC, such as mobile phone apps or compact portable readers.

2. **Near point of care**: These tests can be instrument based, preferably battery operated, thus not requiring any special infrastructure as well and can be placed in health clinics without laboratories. Health care workers with basic technical skills (basic pipetting) not requiring precision can perform these tests. An example is a portable nucleic acid amplification test (NAAT) platform.

3. **Low complexity assays:** These tests require an instrument, can be placed in peripheral laboratories (e.g., microscopy centers) and in some cases health clinics with basic laboratory infrastructure and people with basic technical skills can use it. An example is the GeneXpert 10-color platform.

## C. Evidence-driven decision making

Most TPPs provide product characteristics based on published literature, systematic reviews or the upcoming technologies in the pipeline. These inputs from subject matter experts and invaluable, however, for estimates like accuracy and costs of a product, the TPP discussions were informed by a simulation-based model that explicitly quantified the trade-offs between test access and accuracy for potential new TB diagnostics to be utilised within the diagnostic care cascades of

| Complexity | PoC | Near PoC | Low complexity |
|---|---|---|---|
| Equipment | None ✖ | Maybe, preferably battery operated | Yes ✔ |
| Infrastructure | None ✖ | None ✖ | Basic laboratory requirements (i.e required power supply) but non-specialized laboratory infrastructure |
| HR skill | None or minimal skills | Basic technical skills (basic pipetting, precision not critical) | Basic technical skills (basic pipetting, precision not critical) |

**Fig 1. Three types of tests to be used at different levels of the health system and associated complexity.** HR: human resource; POC: point of care.

South Africa, India, and Kenya. Fig 2 provides the conceptual framework for this model and details have been published [8]. Model outputs were presented to the STDG and final characteristics in the WHO TPP were guided by the evidence-driven estimates.

### D. TPP development process

The standard WHO protocol was followed for this TPP document (Fig 3). Draft TPP was prepared that incorporated the modelling (sensitivity, specificity, costs) and the non-modelling components (target population, settings, operational characteristics etc.). This draft was then presented to the STDG in an online meeting explaining the integration of these components in one document.

The Delphi method was used to help reach agreement on proposed characteristics for TB diagnostic products. All STDG members were invited to participate in this process. We used the Welphi platform (www.welphi.com) to send out the survey. The working document and survey questionnaire consisted of 86 parameters. Delphi comments were taken into consideration and appropriate changes were made before making it available for public comment through the WHO platform.

Revised TPP document was shared on the WHO platform for public comment from 31st July 2023–31st August 2023.

Feedback received from the public comment process was analysed. Quantitative and qualitative assessments from the Delphi step and public comment were discussed at the virtual stakeholder consultation on 13th and 14th September 2023. During this 2-day consultation, results for all 86 characteristics were presented and discussed. Consensus was reached for each characteristic via discussion, and virtual chat to record for any agreements and/or disagreements.

### Ethical consideration

All members of the STDG made a declaration on potential interests, which were reviewed and managed by WHO; a statement on the declarations is available in the WHO TPP document [9]. As member of the WHO TPP Development group,

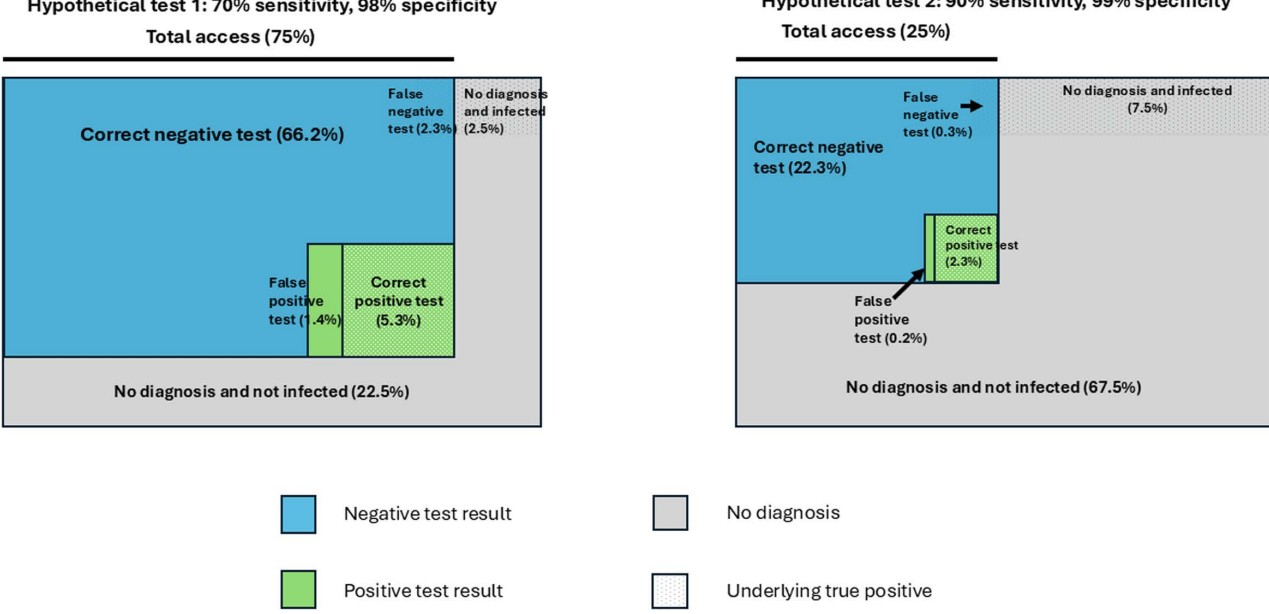

**Fig 2. Trade-off between test accuracy and test access, assuming an underlying disease prevalence of 10%.**

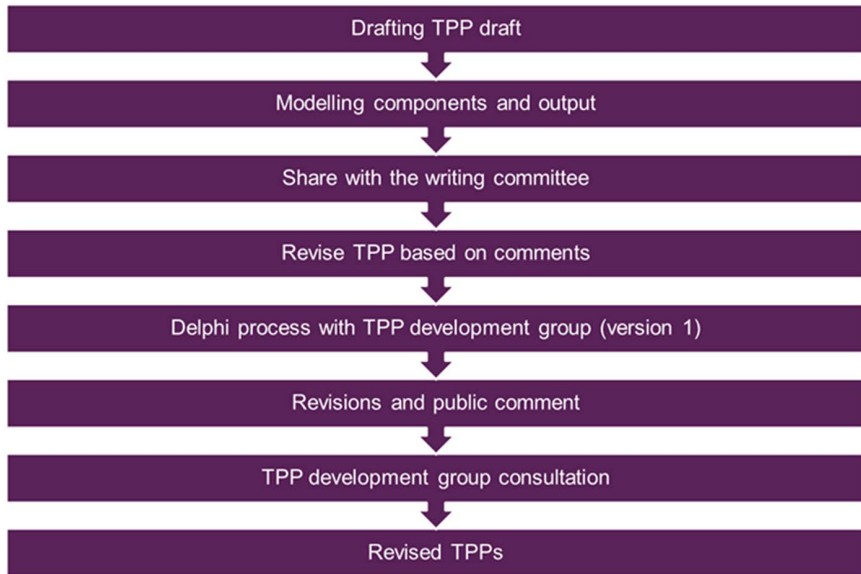

**Fig 3. TPP development workflow.**

and its core task being to provide technical expertise for the TPP document, ethics approval was not needed. These tasks were included in the declaration forms managed by the WHO to obtain informed consent in a written format from the TPP development group. Additionally, during the public comment process, information on this process was shared on the WHO webpage and as a standard practice. This also included getting an approval from the participants to share their comments and personal information for analysis. If the participants did not approve of sharing their responses and personal information, they were not included in the analyses.

## Results

For the Delphi-like survey, 49 (84%) people responded and provided comments on the TPP characteristics. For the public comment step, 205 people consented to share their information and responses to the TPP survey. Fig 4 and Fig 5 provide respondents' information on various constituencies and countries they represent. This consultation took place from 9th June to 10th July 2023.

Results of the Delphi survey are provided in the S1 Table

### Primary updates to the TPP document

This TPP has two major updates to the previously published series of TPP a) it incorporates sputum and non-sputum-based tests under one TPP, and b) using modelling-based approach to assess the acceptable trade-offs between an easily accessible sample (such as urine, swab, breath etc), can be placed as a point of care, near point of care level, with a lower accuracy estimates and its impact. Therefore, this TPP includes characteristics for all tests, being technology agnostic with the main goal of rapidly and accurately detecting TB at peripheral level.

### Scope of TPP

As the primary objective of this document, the new tests should be able to detect TB rapidly and accurately at peripheral level to enable healthcare providers to initiate TB therapy in the same clinical encounter. Optimally, the test(s) should also

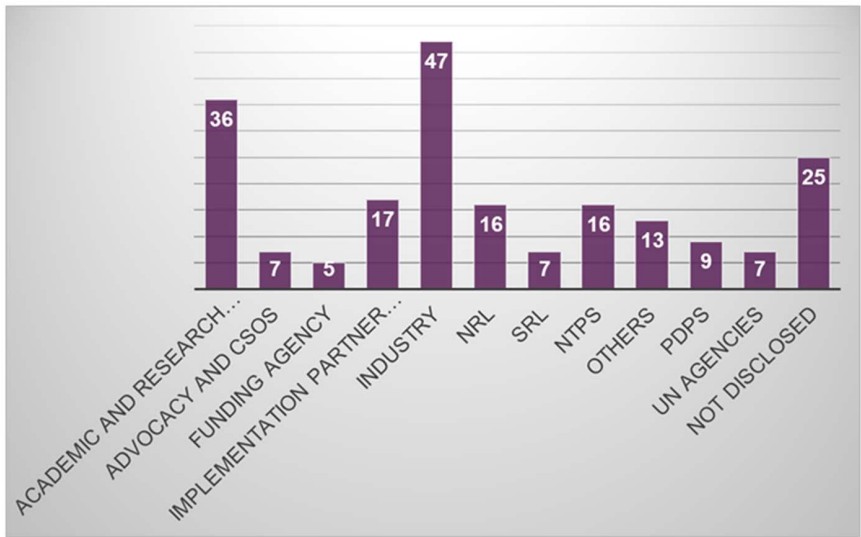

**Fig 4. Respondents' constituency type.** CSOs: Civil Society Organizations; NRL: National Reference Laboratories; SRL: Supranational laboratories; PDPs: Product development partnerships; UN: United Nations.

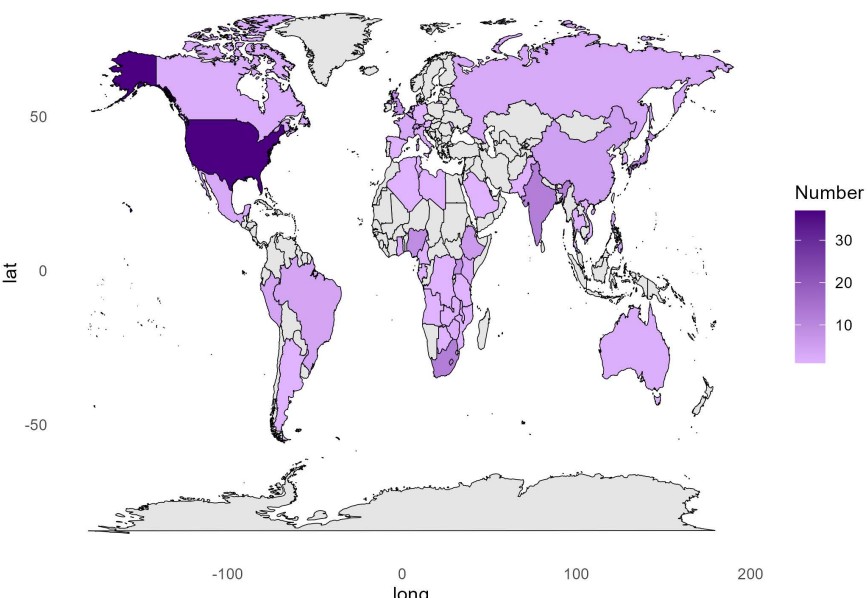

**Fig 5. Respondents' country representation.** This map has been created using "mapdata" package from **R.** Basemap source: CIA World DataBank II, accessed via the mapdata R package (Brownrigg, 2018).This graph shows geographical representation of the respondents from the public comment to the TPP draft. The intensity of purple shading in the graph represents the number of respondents from each country.

be able to detect drug resistance detection and previously published TPP on tests for DST [6] provide guidance on priority ranking and sequence of drugs manufacturers can refer to.

As diagnosis of childhood TB is an important global health need, acknowledging the difficulty in performing validation studies in specimens collected from children, new tests should optimally try to expand their validation studies in children, whenever possible.

Table 1 provides details on the characteristics for scope of this TPP catering to the key objective, target population, setting etc.

## Performance

To evaluate the accuracy estimates for TB diagnostic tests, we used a modelling approach to assess the tradeoffs between test accuracy and increased access to testing and this provided evidence to inform this TPP document. In this model, estimating sensitivity of the test was the primary goal as the group decided that for a diagnostic test, only high specificity tests would be acceptable with >98% specificity when compared to liquid culture.

Access to testing was modelled using specimen type (sputum, non-sputum) and level of healthcare setting (PoC, Non-PoC and low complexity testing). For example, non-sputum-based testing will increase access to testing in people who cannot produce sputum such as children and people living with HIV. This led to six possible combinations and Table 2 provides accuracy estimates for these combinations.

Data from three countries (India, South Africa and Kenya) were used to provide baseline estimates for this model. Details of this work are published elsewhere [8]. Final estimates from this work were discussed during the virtual webinars with the STDG and were slightly modified based on the Delphi process, public comments and the final TPP consultation with the STDG. These estimates did not vary significantly but were modified to reflect ease of understanding and being aspirational yet realistic.

## Cost

To provide estimates for cost per test, we used a cost neutralization modelling approach for cost per person correctly identified, but not including the total cost of resulting downstream consequences of correctly/incorrectly diagnosed. This mathematical simulation model using estimates from South Africa, Kenya and India were also used for driving the discussions with the STDG.

Within the cost-neutralization modelling, there were two different methods we used:

a. Cost neutralization with baseline costs for a sputum-based low-complexity assay of $8 for minimal and $4 for optimal.

b. Cost neutralization with baseline costs for a sputum-based low-complexity assay of $8 for minimal, but increasing the case detection by 30% for optimal

This model used minimum acceptable sensitivity from the modeling work explained above for sensitivity estimates. To achieve the same number of positives identified for a given budget, we then related the test sensitivity identified for each

**Table 1. Scope of Diagnostic TPP for TB detection at peripheral level.**

| Characteristic | Minimal | Optimal |
|---|---|---|
| **Scope** | | |
| **Goal** | To provide the characteristics for a diagnostic test to detect pulmonary TB (PTB), at peripheral level, to support initiation of TB therapy during the same clinical encounter or on the same day in peripheral settings. | To provide the characteristics for a diagnostic test to detect pulmonary TB (PTB), and extrapulmonary (EP-TB) with drug resistance detection at peripheral level, to support initiation of TB therapy during the same clinical encounter in peripheral settings. |
| **Target population** | Adults and adolescents presumed to have pulmonary TB, irrespective of HIV status | Adults, adolescents, and children presumed to have PTB or EPTB disease, irrespective of HIV status |
| **Target user of test** | Health care workers with basic technical skills (non-precision pipetting, minimal sample processing) | Community health workers and/or lay caregivers with minimal training |
| **Setting (level of the Healthcare system)** | Peripheral microscopy centers; Primary health clinics | Primary health clinics without labs; Community level |

respective scenario from the mathematical model to the number of tests that would need to be done to achieve an equivalent number of positives identified for the same budget for the current recommended standard of care (*Standard of care: sputum, low complexity assay, 90% sensitivity, 98% specificity, calculated for a 10% underlying prevalence*). Under this approach, the $8 test was used as the baseline, with $4 as the baseline under optimal conditions.

Another method where under optimal conditions, we estimated that we would expect that new tests will increase access to testing, thus increasing case detection. To account for this increased access to testing under optimal conditions, we estimated costs of the tests if the case detection were to be increased by 30%. Given that we would expect a greater number of tests to be done, the cost per test would need to decrease to achieve cost neutralization. Details of this work have been published elsewhere [8].

Both these methods and their outputs were presented to the STDG for discussions and the final estimates are provided in Table 3. This exercise used the 6 combinations as mentioned in Table 2, however, for costs, STDG decided to only provide cost estimates based on healthcare setting irrespective of the specimen type.

Lastly, for cost of the instrument, minimally, for an instrument based test, the cost should target to be less than $2000 and ideally, it should be an instrument free, point of care test. The lower the capital costs of the instrument are, the lower the initial cost would be, and thus the barrier to implementation would also be lower, particularly since the volume of instruments that would be distributed to peripheral centers is sizeable. The cost of the instrument should be evidence-based and should also include warranties, service contracts and technical support. Novel acquisition models, such as reagent rental or a cost-per-result model should be considered to improve affordability of instruments for low- and middle-income countries.

Table 2. *Performance estimates for TB diagnostic tests*#.

| Characteristic | Minimal | Optimal* |
|---|---|---|
| **Diagnostic sensitivity for TB detection** | | |
| **Sputum, Low complexity assay** | 90% | ≥ 95% |
| **Sputum, Near PoC** | 85% | |
| **Sputum, PoC** | 80% | |
| **Non-sputum, Low complexity assay** | 80% | |
| **Non-sputum, Near PoC** | 75% | |
| **Non-sputum, PoC** | 65% | |
| **Diagnostic Specificity for TB detection** | > 98% for a single test when compared with liquid culture | |

#All estimates provided are with respect to liquid culture serving as the reference standard.

*For optimal estimates, to cater to the evolving innovation in the field of diagnostics, it was decided to keep the estimates as aspirational for all sputum and non-sputum based to a high sensitivity of ≥ 95% when compared to liquid culture as the reference standard.

Table 3. Costs.

| Characteristic | Minimal | Optimal |
|---|---|---|
| **Cost of individual test (reagent costs only; at scale; ex-works)** | | |
| **Low complexity assay** | ≤$8 | ≤ $5 |
| **Near PoC** | ≤$6 | ≤ $4 |
| **PoC** | ≤$4 | ≤ $2 |
| **Capital cost for the instrument** | <US$ 2000 | None (optimally a PoC test) |

## Operational characteristics

For this document, these characteristics were split under two sections: a) sample and equipment requirements and b) data requirements.

For sample and equipment requirements, these parameters are all linked to ensure that specimens are easily collectable at peripheral settings, with an optimal option of enabling self-collectible clinical samples for TB detection. The tests should have a minimal number of steps involved after sample collection to ensure a short time to result.

Parameters like batching, daily throughput of more than 8 tests, easy and safe waste disposal were also considered in this TPP, and details are described in Table 4. Manufacturers should also consider making instruments and tests which have minimal impact on the environment and should ensure minimizing waste, maximizing reusability and use of recyclable materials. To ensure the tests are robust and can be implemented in resource limited settings, the manufacturers should also carefully design their products keeping in mind high temperatures and humidity conditions, with no cold chain required for reagent storage and transport.

To ensure clinical decisions being made in the same clinical encounter, testing on these new platforms also requires connectivity, automated result interpretation etc. All the raw data, trends, error codes, reasons for test failures should be built in within the instrument. Ideally, for an instrument-free PoC test, any ancillary instrument like mobile phones, should have this capability as well. Simple to interpret and display results are important in these settings. Table 4 provides a combination of connectivity interface/channel given the test settings and facility infrastructure vary at peripheral facilities/centers. The full functionality of the test device should not depend on the availability of connectivity ports/solutions. Connectivity of diagnostic devices should allow for the visibility of data for reporting at both local and national levels, which can be used to further improve national programmes.

Manufacturers should be able to provide flexibility in data storage locations to customize it based on the data governance policies of where the test will be implemented. Data security and privacy are important considerations that manufacturers should prioritize with these new test platforms. Additionally, for ease of usability, language support should be provided in one popular language, such as the official language and any language mandated by local regulatory or trade compliance requirements.

## Discussion

The WHO TPPs are intended to support and facilitate new, innovative product development that can help cater to critical public health needs. The 2014 TPPs provided set requirements for specific use-cases of new diagnostic tests and served to inform the development of several new TB diagnostics such as Xpert MTB/RIF, Xpert Ultra, Truenat, Xpert XDR, LF-LAM etc for TB detection. Since then, many new tests have entered the market that include different drug resistance target with increased throughput. There are upcoming technologies including alternative sample types such as swabs [10–16], breath [17–19], cough with artificial intelligence etc [20,21]. TPPs therefore served as important guidance documents to aid communication between various stakeholders by laying out a set of criteria.

To decrease the diagnostic gap and to meet End-TB goals, there is an urgent need to develop, validate and scale up innovative new tools that can be deployed at peripheral settings, diagnosing people who cannot produce sputum such as PLHIV, pediatric population etc. To ensure that these innovative tools are impactful, it is imperative that they are designed and manufactured keeping in mind low resource settings, high temperature or humidity settings, and lack of proper infrastructure and trained laboratory personnel.

This was the first TB diagnostic TPP development process where a model-based approach was used to drive discussions on performance of the new tests and cost per test. This was an important step to have evidence-based decision making in this updated TPP. This approach provided a comprehensive understanding of the interplay between accessibility and test performance in the context of new TB diagnostics, thus providing realistic yet aspirational targets for new

**Table 4. Operational characteristics.**

| Characteristic | Minimal | Optimal |
|---|---|---|
| **Operational characteristics (1): Sample and equipment requirements** | | |
| Sample type | Sputum and/or non-sputum samples which are not more complex to obtain than sputum | Self-collectable clinical specimens |
| Manual preparation of samples (steps needed after obtaining sample) | Up to 3 steps for preprocessing and running the test. No need for precise measuring and sampling | Integrated sample preparation and detection in a closed system with minimal technical input |
| Time to result | Less than 60 minutes | Less than 15 mins |
| Daily throughput | ≥8 tests | |
| Sample capacity and throughput | Multiple samples should be able to be tested at the same time; random access should be possible | |
| Walk-away operation | No more than 2 steps of operator intervention should be needed once the sample has been placed into or on the test/system | No instrument required |
| Biosafety | Requirements are similar to those for smear microscopy (low-risk TB laboratories) | Minimal infectious aerosol risk |
| Waste disposal – solid | Should require no more than current WHO-endorsed TB assays at the peripheral level | Should require less than current rapid molecular tests for TB Reusable, recyclable, or non-plastic alternatives to disposable materials |
| Waste disposal – infectious | Similar to those for smear microscopy (low-risk TB laboratories) | Less than smear microscopy (low-risk TB laboratories) |
| Instrument | For instrument-based tests, build on a modular concept allowing tailoring to meet needs and upgrade additional functionalities at any time | No instrumentation required |
| Power requirements | Standard operating currents with built-in UPS for utilization in locations with variable power. Using battery powered platforms, and/or other forms of renewable energy like solar power would be preferable. | Not applicable |
| Maintenance and calibration | Preventative maintenance @1 year or >1000 samples; include maintenance alert. Need for calibration on-site on a yearly basis by minimally trained technician or instrument should calibrate itself. | None, swap out or replace ancillary devices when needed. Can be calibrated remotely or no calibration is needed. |
| Regulatory requirements | Manufacturing of the assay and system should comply with ISO13485 as well as ISO 14971 or higher standards or regulations, and comply with ISO IEC 62304 (Medical device software — Software life cycle processes); the manufacturing facility should be assessed at a high-risk classification and certified for use by one of the regulatory authorities of the founding members of the International Medical Device Regulators Forum (formerly known as Global Harmonization Task Force); the assay must be registered for in vitro diagnostic use | |
| Operating environment, temperature and humidity level | Between +5°C and +40°C with up to 70% humidity. It is important to adequately protect optics from dust in these settings | Between +5°C and +50°C with up to 90% humidity |
| Reagent kit – transport | No cold chain should be required; Should be able to tolerate stress during transport for at least 72 hours at –15°C to +40°C | No cold chain required; Should be able to tolerate stress during transport for at least 96 hours at –25°C to +50°C |
| Reagent kit – storage and stability | 12 months at +5°C to +35°C with up to 70% humidity; should be able to tolerate stress during transport for at least 72 hours at +40°C; no cold chain should be required | 2 years at +5°C to +40°C with up to 90% humidity; should be able to tolerate stress during transport for at least 72 hours at +50°C; no cold chain should be required |
| Training and education | 1 day for staff with the ability to perform low complexity assays | None or < 1 day for caregivers or community health workers with minimal training |

*(Continued)*

**Table 4.** (Continued)

| Characteristic | Minimal | Optimal |
|---|---|---|
| Environmental Impact | Minimize adverse impact on the environment | Tests and any associated instruments should minimize adverse impact on the environment. This includes the potential to produce tests locally, minimizing waste and maximizing reusability and recycling of by-products, multi-use platforms, recycling of instruments at the end of their life, and low power consumption and radiation emissions |
| **Operational characteristics (2): Data requirements** | | |
| Built-in analytics (for instrument-based tests) | Built-in analytics for instrument and test data; a PC should not be required. | |
| Result documentation, data display | Digital read-outs to display assay details including results screen and the ability to save and export results should be included | Access to assay details, e.g., QR code on a test device or PoC tests to digitally record and report data. |
| Connectivity | All test and device data can be securely transmitted via a standard cable connection interface (USB, ethernet) or wireless connection, including at least one of the following: Bluetooth, Wi-fi, mobile broadband modem (embedded or external), Data from the instruments should be compatible with different information systems at health facility levels using industry standard formats/protocols. | For device-based tests, off-line data storage should be available for data up to 3 months and should be interoperable over W/LAN and with information management systems. Non-device based tests may have ancillary readers and other data capture apps |
| Interoperability Standards and Format | Data, including device usage data, error rates, number of invalid tests, etc. can be exported in standard formats, including but not limited to:<br>• XML<br>• CSV<br>• 3rd party instrument, e.g., USB | Same as minimal plus transmitted data (including results) from devices should be encoded using health information exchange (HIE) standards including, HL7 FHIR. |
| Software/OS Maintenance | As applicable, POC device should allow for routine software/operating system maintenance (automatically or manually) | |
| Data Storage | The administrative institution (MoH or TB programs) of sites where tests are deployed shall be able to specify or agree with the storage location of the device data without affecting the support and optimal use of the device. | |
| Data Ownership | Test data, its management, and ownership must be in compliance with local regulations. | |
| Security and privacy | To facilitate use by health programmes in accordance with the laws, regulations, and policies in their settings and with best practices, the device shall provide configurable features so that personal data can be:<br>a. gathered transparently to users and people who are taking the tests, including consent,<br>b. collected and processed only for purposes compatible with the health programme's purposes,<br>c. limited to what is relevant and necessary,<br>d. collected accurately,<br>e. stored in an identifiable form no longer than necessary and<br>f. secured for integrity and confidentiality, with encryption at rest and in transmission. | |
| Language support | For each country in which the test is deployed, one popular language, such as the official language or de facto national language, and any language mandated by local regulatory or trade compliance requirements | Same as minimal plus additional languages that enable use by additional residents of the location of deployment |

technologies. This work helped in assessing impact of easily accessible tests despite lower accuracy estimates compared to current tests. It is expected to encourage innovative tests to enter the market which can reach more people in need.

This latest TPP is expected to transform the current TB diagnostic landscape from largely laboratory-based sputum testing to non-laboratory-based testing using alternative sample types. Ideally, a true PoC test is needed for TB, and the performance targets set are likely to be achievable in the next 5 years. Despite its lower accuracy estimates and based on modelling, it will likely have the necessary impact in reaching patients with limited access, especially in resource-constrained settings. Major investments are required to see this becoming a reality. In the absence of a true PoC test, low cost near PoC technologies are already reaching market and offer the first a quickest step forward and have great

potential to meet many of the requirements specified in the TPP. However, the TPP is designed to be technology agnostic and alternative approaches and technologies that could meet the targets are strongly encouraged.

TPP documents are a set of useful guidance parameters for product developers and researchers providing a representation of the broader "TB community". These parameters are not to be considered absolute, but indicative to guide the development process. A test may demonstrate excellent impact on patient outcomes leading to a positive policy decision despite not achieving the minimum for all 86 parameters. All new tests need to be validated in appropriate clinical settings demonstrating its important public health benefits for it to be considered for WHO policy recommendations. A combination of robust clinical performance, impact on patient outcomes, and economical and qualitative evidence is usually required for policymaking. Additionally, it is also important to note that it is not just the introduction of a new test that can be a silver bullet, but efficient scale up and implementation is what can help in achieving End TB goals globally.

## Supporting information

**S1 Table. Results of the Delphi survey.**
(XLSX)

## Acknowledgments

We would like to thank the following members of the Scientific TPP Development Group for their support and inputs in finalizing this as a WHO document: Chukwuma Anyaike (National Tuberculosis [TB], Leprosy and Buruli Ulcer Control Programme, Nigeria), Helen Ayles (London School of Hygiene and & Tropical Medicine [LSHTM], United Kingdom of Great Britain and Northern Ireland [United Kingdom]), Ramon Basilio (National TB Reference Laboratory, Research Institute for Tropical Medicine, Philippines), David Branigan (Treatment Action Group, United States of America [USA]), Adithya Cattamanchi (University of California, San Francisco, USA), Daniela Maria Cirillo (WHO Collaborating Centre and TB Supranational Reference Laboratory, San Raffaele Scientific Institute, Italy), Frank Cobelens (University of Amsterdam, Netherlands (Kingdom of the)), Claudia Denkinger (University of Heidelberg, Germany), David Dowdy (Johns Hopkins Bloomberg School of Public Health, USA), Petra de Haas (KNCV Tuberculosis Foundation, Netherlands (Kingdom of the)), Patricia Hall (Centers for Disease Control and Prevention, USA), Rumina Hasan (Aga Khan University, Pakistan), Cathy Hewison (Médecins Sans Frontières, France), Jamilya Ismailova (Abt Associates, Civil Society Representative, Tajikistan), Davaalkham Jagdagsuren (Mongolia National Centre for Communicable Diseases, Mongolia), Rajendra Panduranga Joshi (Ministry of Health and Family Welfare, India), Gulmira Kalmambetova (Ministry of Health, Kyrgyzstan), Jacqueline Kisia (National TB Programme, Kenya), Katharina Kranzer (LSHTM, United Kingdom), Rhea Lobo (Independent Health Journalist, Denmark), Peter MacPherson (University of Glasgow, United Kingdom), Sandeep Meharwal (FHI 360 Asia Pacific Regional Office, Thailand), Paolo Miotto (WHO Collaborating Centre and TB Supranational Reference Laboratory, San Raffaele Scientific Institute, Italy), Troy Murrell (Clinton Health Access Initiative, USA), Ruvandhi Nathavitharana (Harvard Medical School, USA), Norbert Ndjeka (Department of Health of South Africa, South Africa), Van Hung Nguyen (National TB Reference Laboratory, National Lung Hospital, Viet Nam), Mark Nicol (University of Western Australia, Australia), Rustam Nurov (National TB Programme, Tajikistan), Shaheed Vally Omar (National Institute for Communicable Diseases, WHO Supranational TB Reference Laboratory, South Africa), Madhukar Pai (McGill University, Canada), Tiffany Tiara Pakasi (National TB Programme, Indonesia), Paulo Redner (National TB Reference Laboratory, Oswaldo Cruz Foundation, Brazil), Andriansjah Rukmana (University of Indonesia, Indonesia), Anastasia Samoilova (National Medical Research Centre on Phthisiopulmonology and Infectious Diseases, Russian Federation), Mahafuzer Rahman Sarker (TB–Leprosy and AIDS/STD programme, Bangladesh), Siva Kumar Shanmugam (National Institute for Research in Tuberculosis, India), Thomas Shinnick (Independent Laboratory Consultant, USA), Nicole de Souza (National TB Programme, Brazil), Willy Ssengooba (Makerere University, Uganda), Sabira Tahseen (National TB

Reference Laboratory, Pakistan), Diana Vakhrusheva (Ministry of Health, Russian Federation), Dinh Van Luong (National Lung Hospital, Viet Nam) and Zhao Yanlin (National Clinical Centre on Tuberculosis, China).

## Author contributions

**Conceptualization:** Mikashmi Kohli, Nazir Ismail, Matteo Zignol, Tereza Kasaeva, Puneet Dewan, Morten Ruhwald.

**Data curation:** Mikashmi Kohli, Morten Ruhwald.

**Formal analysis:** Mikashmi Kohli.

**Funding acquisition:** Mikashmi Kohli, Morten Ruhwald.

**Investigation:** Mikashmi Kohli, Morten Ruhwald.

**Methodology:** Mikashmi Kohli, Alexei Korobitsyn, Nazir Ismail, Morten Ruhwald.

**Project administration:** Mikashmi Kohli, Alexei Korobitsyn.

**Resources:** Mikashmi Kohli, Alexei Korobitsyn, Nazir Ismail, Matteo Zignol, Tereza Kasaeva, Morten Ruhwald.

**Software:** Mikashmi Kohli.

**Supervision:** Nazir Ismail, Matteo Zignol, Tereza Kasaeva, Morten Ruhwald.

**Validation:** Mikashmi Kohli, Alexei Korobitsyn, Nazir Ismail, Matteo Zignol, Tereza Kasaeva, Morten Ruhwald.

**Visualization:** Mikashmi Kohli, Nazir Ismail.

**Writing – original draft:** Mikashmi Kohli.

**Writing – review & editing:** Mikashmi Kohli, Alexei Korobitsyn, Nazir Ismail, Matteo Zignol, Tereza Kasaeva, Puneet Dewan, Morten Ruhwald.

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
