## [Decision Letter · Decision Letter 0]

PGPH-D-25-00372

WHO target product profile for TB detection at peripheral settings: 2024 update

Dear Dr. Kohli,

Thank you for submitting your manuscript to PLOS Global Public Health. After careful consideration, we feel that it has merit but does not fully meet PLOS Global Public Health’s publication criteria as it currently stands. Therefore, we invite you to submit a revised version of the manuscript that addresses the points raised during the review process.

We look forward to receiving your revised manuscript.

Kind regards,

Sadia Shakoor

Academic Editor

Journal Requirements:

1. Please provide additional details regarding participant consent. In the ethics statement in the Methods and online submission information, please ensure that you have specified (1) whether consent was informed and (2) what type you obtained (for instance, written or verbal, and if verbal, how it was documented and witnessed). If your study included minors, state whether you obtained consent from parents or guardians. If the need for consent was waived by the ethics committee, please include this information. If you are reporting a retrospective study of medical records or archived samples, please ensure that you have discussed whether all data were fully anonymized before you accessed them and/or whether the IRB or ethics committee waived the requirement for informed consent. If patients provided informed written consent to have data from their medical records used in research, please include this information. 2. Please provide separate figure files in .tif or .eps format. For more information about figure files please see our guidelines:  LINK https://journals.plos.org/globalpublichealth/s/figures https://journals.plos.org/globalpublichealth/s/figures#loc-file-requirements 3. Please upload a copy of Figures 1, 2, 3, 4a and 4b which you refer to in your text on page 5,6, 7, 8. Or, if the figure is no longer to be included as part of the submission please remove all reference to it within the text. 4. In the online submission form, you indicated that All data has been provided as part of this manuscript and any additional information can be provided on request as well.  All PLOS journals now require all data underlying the findings described in their manuscript to be freely available to other researchers, either 1. In a public repository, 2. Within the manuscript itself, or 3. Uploaded as supplementary information. This policy applies to all data except where public deposition would breach compliance with the protocol approved by your research ethics board. If your data cannot be made publicly available for ethical or legal reasons (e.g., public availability would compromise patient privacy), please explain your reasons by return email and your exemption request will be escalated to the editor for approval. Your exemption request will be handled independently and will not hold up the peer review process, but will need to be resolved should your manuscript be accepted for publication. One of the Editorial team will then be in touch if there are any issues. 5. Figure 4b: please (a) provide a direct link to the base layer of the map (i.e., the country or region border shape) and ensure this is also included in the figure legend; and (b) provide a link to the terms of use / license information for the base layer image or shapefile. We cannot publish proprietary or copyrighted maps (e.g. Google Maps, Mapquest) and the terms of use for your map base layer must be compatible with our CC-BY 4.0 license.  Note: if you created the map in a software program like R or ArcGIS, please locate and indicate the source of the basemap shapefile onto which data has been plotted. If your map was obtained from a copyrighted source please amend the figure so that the base map used is from an openly available source. Alternatively, please provide explicit written permission from the copyright holder granting you the right to publish the material under our CC-BY 4.0 license. Please note that the following CC BY licenses are compatible with PLOS license: CC BY 4.0, CC BY 2.0 and CC BY 3.0, meanwhile such licenses as CC BY-ND 3.0 and others are not compatible due to additional restrictions.  If you are unsure whether you can use a map or not, please do reach out and we will be able to help you. The following websites are good examples of where you can source open access or public domain maps: * U.S. Geological Survey (USGS) - All maps are in the public domain. (http://www.usgs.gov) * PlaniGlobe - All maps are published under a Creative Commons license so please cite “PlaniGlobe, http://www.planiglobe.com, CC BY 2.0” in the image credit after the caption. (http://www.planiglobe.com/?lang=enl) * Natural Earth - All maps are public domain. (http://www.naturalearthdata.com/about/terms-of-use/)

Additional Editor Comments (if provided):

Reviewers' comments:

Reviewer's Responses to Questions

**Comments to the Author**

1. Does this manuscript meet PLOS Global Public Health’s publication criteria ? Is the manuscript technically sound, and do the data support the conclusions? The manuscript must describe methodologically and ethically rigorous research with conclusions that are appropriately drawn based on the data presented.

Reviewer #1: Yes

Reviewer #2: Yes

2. Has the statistical analysis been performed appropriately and rigorously?

Reviewer #1: Yes

Reviewer #2: N/A

3. Have the authors made all data underlying the findings in their manuscript fully available (please refer to the Data Availability Statement at the start of the manuscript PDF file)?

Reviewer #1: Yes

Reviewer #2: Yes

4. Is the manuscript presented in an intelligible fashion and written in standard English?

Reviewer #1: Yes

Reviewer #2: Yes

5. Review Comments to the Author

Reviewer #1: Clear and accurate description of the process to develop WHO target product profiles. A reference to the WHO TPP Directory could have been included: https://www.who.int/our-work/science-division/research-for-health/target-product-profile-directory and I'd ensure that the original TPP is uploaded in the Directory as it is an additional way to disseminate this product to relevant audience members.

Reviewer #2: This is a well-written manuscript about Target Product Profiles (TPPs) in TB diagnostics. It highlights:

-The evolution of TB diagnostics

-The importance of accessibility

-The shift towards non-laboratory-based tests and the push for point-of-care (PoC) testing.

- The role of modeling in TPP development, balancing performance, cost, and real-world feasibility.

and most importantly how approval should be given for tests even for tests meeting minimal criteria to improve access and availability.

A few minor comments:

1. Figure 4b) is not clear. Their should be a brief explanation with the figure.

2. There are limited respondents from Africa and South East Asia. Would the data provided accurately represent these regions, especially in terms of pricing of tests ?

3. It would be good to mention the various tests considered in sputum and non-sputum POC, near POC and low complexity assays.

6. PLOS authors have the option to publish the peer review history of their article (what does this mean? ). If published, this will include your full peer review and any attached files.

**Do you want your identity to be public for this peer review?** For information about this choice, including consent withdrawal, please see our Privacy Policy .

Reviewer #1: **Yes: ** Mercedes Perez Gonzalez

Reviewer #2: No

---

## [Editor Report · Decision Letter 1]

WHO target product profile for TB detection at peripheral settings: 2024 update

PGPH-D-25-00372R1

Dear Dr. Kohli,

We are pleased to inform you that your manuscript 'WHO target product profile for TB detection at peripheral settings: 2024 update' has been provisionally accepted for publication in PLOS Global Public Health.

Best regards,

Sadia Shakoor

Academic Editor